# Minimizing Quadratic Functions in Constant Time

**Kohei Hayashi**
National Institute of Advanced Industrial Science and Technology
hayashi.kohei@gmail.com

**Yuichi Yoshida**
National Institute of Informatics *and* Preferred Infrastructure, Inc.
yyoshida@nii.ac.jp

## Abstract

A sampling-based optimization method for quadratic functions is proposed. Our method approximately solves the following $n$-dimensional quadratic minimization problem in constant time, which is independent of $n$: $z^* = \min_{\boldsymbol{v} \in \mathbb{R}^n} \langle \boldsymbol{v}, A\boldsymbol{v} \rangle + n \langle \boldsymbol{v}, \mathrm{diag}(\boldsymbol{d})\boldsymbol{v} \rangle + n \langle \boldsymbol{b}, \boldsymbol{v} \rangle$, where $A \in \mathbb{R}^{n \times n}$ is a matrix and $\boldsymbol{d}, \boldsymbol{b} \in \mathbb{R}^n$ are vectors. Our theoretical analysis specifies the number of samples $k(\delta, \epsilon)$ such that the approximated solution $z$ satisfies $|z - z^*| = O(\epsilon n^2)$ with probability $1 - \delta$. The empirical performance (accuracy and runtime) is positively confirmed by numerical experiments.

## 1 Introduction

A quadratic function is one of the most important function classes in machine learning, statistics, and data mining. Many fundamental problems such as linear regression, $k$-means clustering, principal component analysis, support vector machines, and kernel methods [14] can be formulated as a minimization problem of a quadratic function.

In some applications, it is sufficient to compute the minimum value of a quadratic function rather than its solution. For example, Yamada *et al.* [21] proposed an efficient method for estimating the Pearson divergence, which provides useful information about data, such as the density ratio [18]. They formulated the estimation problem as the minimization of a squared loss and showed that the Pearson divergence can be estimated from the minimum value. The least-squares mutual information [19] is another example that can be computed in a similar manner.

Despite its importance, the minimization of a quadratic function has the issue of scalability. Let $n \in \mathbb{N}$ be the number of variables (the "dimension" of the problem). In general, such a minimization problem can be solved by quadratic programming (QP), which requires $\mathrm{poly}(n)$ time. If the problem is convex and there are no constraints, then the problem is reduced to solving a system of linear equations, which requires $O(n^3)$ time. Both methods easily become infeasible, even for medium-scale problems, say, $n > 10000$.

Although several techniques have been proposed to accelerate quadratic function minimization, they require at least linear time in $n$. This is problematic when handling problems with an ultrahigh dimension, for which even linear time is slow or prohibitive. For example, stochastic gradient descent (SGD) is an optimization method that is widely used for large-scale problems. A nice property of this method is that, if the objective function is strongly convex, it outputs a point that is sufficiently close to an optimal solution after a constant number of iterations [5]. Nevertheless, in each iteration, we need at least $O(n)$ time to access the variables. Another technique is low-rank approximation such as Nyström's method [20]. The underlying idea is the approximation of the problem by using a low-rank matrix, and by doing so, we can drastically reduce the time

complexity. However, we still need to compute the matrix–vector product of size $n$, which requires $O(n)$ time. Clarkson *et al.* [7] proposed sublinear-time algorithms for special cases of quadratic function minimization. However, it is "sublinear" with respect to the number of pairwise interactions of the variables, which is $O(n^2)$, and their algorithms require $O(n \log^c n)$ time for some $c \geq 1$.

**Our contributions:** Let $A \in \mathbb{R}^{n \times n}$ be a matrix and $\boldsymbol{d}, \boldsymbol{b} \in \mathbb{R}^n$ be vectors. Then, we consider the following quadratic problem:

$$\underset{\boldsymbol{v} \in \mathbb{R}^n}{\text{minimize}} \; p_{n,A,\boldsymbol{d},\boldsymbol{b}}(\boldsymbol{v}), \text{ where } p_{n,A,\boldsymbol{d},\boldsymbol{b}}(\boldsymbol{v}) = \langle \boldsymbol{v}, A\boldsymbol{v} \rangle + n\langle \boldsymbol{v}, \text{diag}(\boldsymbol{d})\boldsymbol{v} \rangle + n\langle \boldsymbol{b}, \boldsymbol{v} \rangle. \quad (1)$$

Here, $\langle \cdot, \cdot \rangle$ denotes the inner product and $\text{diag}(\boldsymbol{d})$ denotes the matrix whose diagonal entries are specified by $\boldsymbol{d}$. Note that a constant term can be included in (1); however, it is irrelevant when optimizing (1), and hence we ignore it.

Let $z^* \in \mathbb{R}$ be the optimal value of (1) and let $\epsilon, \delta \in (0, 1)$ be parameters. Then, the main goal of this paper is the computation of $z$ with $|z - z^*| = O(\epsilon n^2)$ with probability at least $1 - \delta$ in *constant time*, that is, independent of $n$. Here, we assume the real RAM model [6], in which we can perform basic algebraic operations on real numbers in one step. Moreover, we assume that we have query accesses to $A$, $\boldsymbol{b}$, and $\boldsymbol{d}$, with which we can obtain an entry of them by specifying an index. We note that $\boldsymbol{z}^*$ is typically $\Theta(n^2)$ because $\langle \boldsymbol{v}, A\boldsymbol{v} \rangle$ consists of $\Theta(n^2)$ terms, and $\langle \boldsymbol{v}, \text{diag}(\boldsymbol{d})\boldsymbol{v} \rangle$ and $\langle \boldsymbol{b}, \boldsymbol{v} \rangle$ consist of $\Theta(n)$ terms. Hence, we can regard the error of $\Theta(\epsilon n^2)$ as an error of $\Theta(\epsilon)$ for each term, which is reasonably small in typical situations.

Let $\cdot|_S$ be an operator that extracts a submatrix (or subvector) specified by an index set $S \subset \mathbb{N}$; then, our algorithm is defined as follows, where the parameter $k := k(\epsilon, \delta)$ will be determined later.

---
**Algorithm 1**

---
**Input:** An integer $n \in \mathbb{N}$, query accesses to the matrix $A \in \mathbb{R}^{n \times n}$ and to the vectors $\boldsymbol{d}, \boldsymbol{b} \in \mathbb{R}^n$, and $\epsilon, \delta > 0$
1: $S \leftarrow$ a sequence of $k = k(\epsilon, \delta)$ indices independently and uniformly sampled from $\{1, 2, \ldots, n\}$.
2: **return** $\frac{n^2}{k^2} \min_{\boldsymbol{v} \in \mathbb{R}^n} p_{k,A|_S,\boldsymbol{d}|_S,\boldsymbol{b}|_S}(\boldsymbol{v})$.

---

In other words, we sample a constant number of indices from the set $\{1, 2, \ldots, n\}$, and then solve the problem (1) restricted to these indices. Note that the number of queries and the time complexity are $O(k^2)$ and $\text{poly}(k)$, respectively. In order to analyze the difference between the optimal values of $p_{n,A,\boldsymbol{d},\boldsymbol{b}}$ and $p_{k,A|_S,\boldsymbol{d}|_S,\boldsymbol{b}|_S}$, we want to measure the "distances" between $A$ and $A|_S$, $\boldsymbol{d}$ and $\boldsymbol{d}|_S$, and $\boldsymbol{b}$ and $\boldsymbol{b}|_S$, and want to show them small. To this end, we exploit graph limit theory, initiated by Lovász and Szegedy [11] (refer to [10] for a book), in which we measure the distance between two graphs on different number of vertices by considering continuous versions. Although the primary interest of graph limit theory is graphs, we can extend the argument to analyze matrices and vectors.

Using synthetic and real settings, we demonstrate that our method is orders of magnitude faster than standard polynomial-time algorithms and that the accuracy of our method is sufficiently high.

**Related work:** Several constant-time approximation algorithms are known for combinatorial optimization problems such as the max cut problem on dense graphs [8, 13], constraint satisfaction problems [1, 22], and the vertex cover problem [15, 16, 25]. However, as far as we know, no such algorithm is known for continuous optimization problems.

A related notion is property testing [9, 17], which aims to design constant-time algorithms that distinguish inputs satisfying some predetermined property from inputs that are "far" from satisfying it. Characterizations of constant-time testable properties are known for the properties of a dense graph [2, 3] and the affine-invariant properties of a function on a finite field [23, 24].

## 2 Preliminaries

For an integer $n$, let $[n]$ denote the set $\{1, 2, \ldots, n\}$. The notation $a = b \pm c$ means that $b - c \leq a \leq b + c$. In this paper, we only consider functions and sets that are measurable.

Let $S = (x_1, \ldots, x_k)$ be a sequence of $k$ indices in $[n]$. For a vector $\boldsymbol{v} \in \mathbb{R}^n$, we denote the *restriction* of $\boldsymbol{v}$ to $S$ by $\boldsymbol{v}|_S \in \mathbb{R}^k$; that is, $(\boldsymbol{v}|_S)_i = v_{x_i}$ for every $i \in [k]$. For the matrix $A \in \mathbb{R}^{n \times n}$, we denote the *restriction* of $A$ to $S$ by $A|_S \in \mathbb{R}^{k \times k}$; that is, $(A|_S)_{ij} = A_{x_i x_j}$ for every $i, j \in [k]$.

## 2.1 Dikernels

Following [12], we call a (measurable) function $f : [0,1]^2 \to \mathbb{R}$ a *dikernel*. A dikernel is a generalization of a *graphon* [11], which is symmetric and whose range is bounded in $[0,1]$. We can regard a dikernel as a matrix whose index is specified by a real value in $[0,1]$. We stress that the term dikernel has nothing to do with kernel methods.

For two functions $f, g : [0,1] \to \mathbb{R}$, we define their inner product as $\langle f, g \rangle = \int_0^1 f(x)g(x)\mathrm{d}x$. For a dikernel $W : [0,1]^2 \to \mathbb{R}$ and a function $f : [0,1] \to \mathbb{R}$, we define a function $Wf : [0,1] \to \mathbb{R}$ as $(Wf)(x) = \langle W(x, \cdot), f \rangle$.

Let $W : [0,1]^2 \to \mathbb{R}$ be a dikernel. The $L_p$ *norm* $\|W\|_p$ for $p \geq 1$ and the *cut norm* $\|W\|_\square$ of $W$ are defined as $\|W\|_p = \left( \int_0^1 \int_0^1 |W(x,y)|^p \mathrm{d}x\mathrm{d}y \right)^{1/p}$ and $\|W\|_\square = \sup_{S,T \subseteq [0,1]} \left| \int_S \int_T W(x,y)\mathrm{d}x\mathrm{d}y \right|$, respectively, where the supremum is over all pairs of subsets. We note that these norms satisfy the triangle inequalities and $\|W\|_\square \leq \|W\|_1$.

Let $\lambda$ be a Lebesgue measure. A map $\pi : [0,1] \to [0,1]$ is said to be *measure-preserving*, if the pre-image $\pi^{-1}(X)$ is measurable for every measurable set $X$, and $\lambda(\pi^{-1}(X)) = \lambda(X)$. A *measure-preserving bijection* is a measure-preserving map whose inverse map exists and is also measurable (and then also measure-preserving). For a measure preserving bijection $\pi : [0,1] \to [0,1]$ and a dikernel $W : [0,1]^2 \to \mathbb{R}$, we define the dikernel $\pi(W) : [0,1]^2 \to \mathbb{R}$ as $\pi(W)(x,y) = W(\pi(x), \pi(y))$.

## 2.2 Matrices and Dikernels

Let $W : [0,1]^2 \to \mathbb{R}$ be a dikernel and $S = (x_1, \ldots, x_k)$ be a sequence of elements in $[0,1]$. Then, we define the matrix $W|_S \in \mathbb{R}^{k \times k}$ so that $(W|_S)_{ij} = W(x_i, x_j)$.

We can construct the dikernel $\widehat{A} : [0,1]^2 \to \mathbb{R}$ from the matrix $A \in \mathbb{R}^{n \times n}$ as follows. Let $I_1 = [0, \frac{1}{n}], I_2 = (\frac{1}{n}, \frac{2}{n}], \ldots, I_n = (\frac{n-1}{n}, \ldots, 1]$. For $x \in [0,1]$, we define $i_n(x) \in [n]$ as a unique integer such that $x \in I_i$. Then, we define $\widehat{A}(x,y) = A_{i_n(x)i_n(y)}$. The main motivation for creating a dikernel from a matrix is that, by doing so, we can define the distance between two matrices $A$ and $B$ of different sizes via the cut norm, that is, $\|\widehat{A} - \widehat{B}\|_\square$.

We note that the distribution of $A|_S$, where $S$ is a sequence of $k$ indices that are uniformly and independently sampled from $[n]$ exactly matches the distribution of $\widehat{A}|_S$, where $S$ is a sequence of $k$ elements that are uniformly and independently sampled from $[0,1]$.

## 3 Sampling Theorem and the Properties of the Cut Norm

In this section, we prove the following theorem, which states that, given a sequence of dikernels $W^1, \ldots, W^T : [0,1]^2 \to [-L, L]$, we can obtain a good approximation to them by sampling a sequence of a small number of elements in $[0,1]$. Formally, we prove the following:

**Theorem 3.1.** *Let $W^1, \ldots, W^T : [0,1]^2 \to [-L, L]$ be dikernels. Let $S$ be a sequence of $k$ elements uniformly and independently sampled from $[0,1]$. Then, with a probability of at least $1 - \exp(-\Omega(kT/\log_2 k))$, there exists a measure-preserving bijection $\pi : [0,1] \to [0,1]$ such that, for any functions $f, g : [0,1] \to [-K, K]$ and $t \in [T]$, we have*

$$|\langle f, W^t g \rangle - \langle f, \pi(\widehat{W^t|_S})g \rangle| = O\left( LK^2 \sqrt{T/\log_2 k} \right).$$

We start with the following lemma, which states that, if a dikernel $W : [0,1]^2 \to \mathbb{R}$ has a small cut norm, then $\langle f, Wf \rangle$ is negligible no matter what $f$ is. Hence, we can focus on the cut norm when proving Theorem 3.1.

**Lemma 3.2.** *Let $\epsilon \geq 0$ and $W : [0,1]^2 \to \mathbb{R}$ be a dikernel with $\|W\|_\square \leq \epsilon$. Then, for any functions $f, g : [0,1] \to [-K, K]$, we have $|\langle f, Wg \rangle| \leq \epsilon K^2$.*

*Proof.* For $\tau \in \mathbb{R}$ and the function $h : [0,1] \to \mathbb{R}$, let $L_\tau(h) := \{x \in [0,1] \mid h(x) = \tau\}$ be the level set of $h$ at $\tau$. For $f' = f/K$ and $g' = g/K$, we have

$$
|\langle f, Wg \rangle| = K^2 |\langle f', Wg' \rangle| = K^2 \Big| \int_{-1}^{1} \int_{-1}^{1} \tau_1 \tau_2 \int_{L_{\tau_1}(f')} \int_{L_{\tau_2}(g')} W(x,y) \mathrm{d}x \mathrm{d}y \mathrm{d}\tau_1 \mathrm{d}\tau_2 \Big|
$$

$$
\leq K^2 \int_{-1}^{1} \int_{-1}^{1} |\tau_1||\tau_2| \left| \int_{L_{\tau_1}(f')} \int_{L_{\tau_2}(g')} W(x,y) \mathrm{d}x \mathrm{d}y \right| \mathrm{d}\tau_1 \mathrm{d}\tau_2
$$

$$
\leq \epsilon K^2 \int_{-1}^{1} \int_{-1}^{1} |\tau_1||\tau_2| \mathrm{d}\tau_1 \mathrm{d}\tau_2 = \epsilon K^2. \qquad \square
$$

To introduce the next technical tool, we need several definitions. We say that the partition $\mathcal{Q}$ is a *refinement* of the partition $\mathcal{P} = (V_1, \ldots, V_p)$ if $\mathcal{Q}$ is obtained by splitting each set $V_i$ into one or more parts. The partition $\mathcal{P} = (V_1, \ldots, V_p)$ of the interval $[0,1]$ is called an *equipartition* if $\lambda(V_i) = 1/p$ for every $i \in [p]$. For the dikernel $W : [0,1]^2 \to \mathbb{R}$ and the equipartition $\mathcal{P} = (V_1, \ldots, V_p)$ of $[0,1]$, we define $W_\mathcal{P} : [0,1]^2 \to \mathbb{R}$ as the function obtained by averaging each $V_i \times V_j$ for $i, j \in [p]$. More formally, we define

$$
W_\mathcal{P}(x,y) = \frac{1}{\lambda(V_i)\lambda(V_j)} \int_{V_i \times V_j} W(x',y') \mathrm{d}x' \mathrm{d}y' = p^2 \int_{V_i \times V_j} W(x',y') \mathrm{d}x' \mathrm{d}y',
$$

where $i$ and $j$ are unique indices such that $x \in V_i$ and $y \in V_j$, respectively.

The following lemma states that any function $W : [0,1]^2 \to \mathbb{R}$ can be well approximated by $W_\mathcal{P}$ for the equipartition $\mathcal{P}$ into a small number of parts.

**Lemma 3.3** (Weak regularity lemma for functions on $[0,1]^2$ [8])**.** *Let $\mathcal{P}$ be an equipartition of $[0,1]$ into $k$ sets. Then, for any dikernel $W : [0,1]^2 \to \mathbb{R}$ and $\epsilon > 0$, there exists a refinement $\mathcal{Q}$ of $\mathcal{P}$ with $|\mathcal{Q}| \leq k2^{C/\epsilon^2}$ for some constant $C > 0$ such that*

$$
\|W - W_\mathcal{Q}\|_\square \leq \epsilon \|W\|_2.
$$

**Corollary 3.4.** *Let $W^1, \ldots, W^T : [0,1]^2 \to \mathbb{R}$ be dikernels. Then, for any $\epsilon > 0$, there exists an equipartition $\mathcal{P}$ into $|\mathcal{P}| \leq 2^{CT/\epsilon^2}$ parts for some constant $C > 0$ such that, for every $t \in [T]$,*

$$
\|W^t - W_\mathcal{P}^t\|_\square \leq \epsilon \|W^t\|_2.
$$

*Proof.* Let $\mathcal{P}^0$ be a trivial partition, that is, a partition consisting of a single part $[n]$. Then, for each $t \in [T]$, we iteratively apply Lemma 3.3 with $\mathcal{P}^{t-1}$, $W^t$, and $\epsilon$, and we obtain the partition $\mathcal{P}^t$ into at most $|\mathcal{P}^{t-1}|2^{C/\epsilon^2}$ parts such that $\|W^t - W_{\mathcal{P}^t}^t\|_\square \leq \epsilon \|W^t\|_2$. Since $\mathcal{P}^t$ is a refinement of $\mathcal{P}^{t-1}$, we have $\|W^i - W_{\mathcal{P}^t}^i\|_\square \leq \|W^i - W_{\mathcal{P}^{t-1}}^i\|_\square$ for every $i \in [t-1]$. Then, $\mathcal{P}^T$ satisfies the desired property with $|\mathcal{P}^T| \leq (2^{C/\epsilon^2})^T = 2^{CT/\epsilon^2}$. $\qquad \square$

As long as $S$ is sufficiently large, $W$ and $\widehat{W}|_S$ are close in the cut norm:

**Lemma 3.5** ((4.15) of [4])**.** *Let $W : [0,1]^2 \to [-L, L]$ be a dikernel and $S$ be a sequence of $k$ elements uniformly and independently sampled from $[0,1]$. Then, we have*

$$
-\frac{2L}{k} \leq \mathbf{E}_S \|\widehat{W}|_S\|_\square - \|W\|_\square < \frac{8L}{k^{1/4}}.
$$

Finally, we need the following concentration inequality.

**Lemma 3.6** (Azuma's inequality)**.** *Let $(\Omega, A, P)$ be a probability space, $k$ be a positive integer, and $C > 0$. Let $\boldsymbol{z} = (z_1, \ldots, z_k)$, where $z_1, \ldots, z_k$ are independent random variables, and $z_i$ takes values in some measure space $(\Omega_i, A_i)$. Let $f : \Omega_1 \times \cdots \times \Omega_k \to \mathbb{R}$ be a function. Suppose that $|f(\boldsymbol{x}) - f(\boldsymbol{y})| \leq C$ whenever $\boldsymbol{x}$ and $\boldsymbol{y}$ only differ in one coordinate. Then*

$$
\Pr\Big[ |f(\boldsymbol{z}) - \mathbf{E}_{\boldsymbol{z}}[f(\boldsymbol{z})]| > \lambda C \Big] < 2e^{-\lambda^2/2k}.
$$

Now we prove the counterpart of Theorem 3.1 for the cut norm.

**Lemma 3.7.** *Let* $W^1, \ldots, W^T : [0,1]^2 \to [-L, L]$ *be dikernels. Let* $S$ *be a sequence of* $k$ *elements uniformly and independently sampled from* $[0,1]$. *Then, with a probability of at least* $1 - \exp(-\Omega(kT/\log_2 k))$, *there exists a measure-preserving bijection* $\pi : [0,1] \to [0,1]$ *such that, for every* $t \in [T]$, *we have*

$$\|W^t - \pi(\widehat{W^t|_S})\|_\square = O\left(L\sqrt{T/\log_2 k}\right).$$

*Proof.* First, we bound the expectations and then prove their concentrations. We apply Corollary 3.4 to $W^1, \ldots, W^T$ and $\epsilon$, and let $\mathcal{P} = (V_1, \ldots, V_p)$ be the obtained partition with $p \leq 2^{CT/\epsilon^2}$ parts such that

$$\|W^t - W^t_\mathcal{P}\|_\square \leq \epsilon L.$$

for every $t \in [T]$. By Lemma 3.5, for every $t \in [T]$, we have

$$\mathbf{E}_S \|\widehat{W^t_\mathcal{P}|_S} - \widehat{W^t|_S}\|_\square = \mathbf{E}_S\|\widehat{(W^t_\mathcal{P} - W^t)|_S}\|_\square \leq \epsilon L + \frac{8L}{k^{1/4}}.$$

Then, for any measure-preserving bijection $\pi : [0,1] \to [0,1]$ and $t \in [T]$, we have

$$\mathbf{E}_S\|W^t - \pi(\widehat{W^t|_S})\|_\square \leq \|W^t - W^t_\mathcal{P}\|_\square + \mathbf{E}_S\|W^t_\mathcal{P} - \pi(\widehat{W^t_\mathcal{P}|_S})\|_\square + \mathbf{E}_S\|\pi(\widehat{W^t_\mathcal{P}|_S}) - \pi(\widehat{W^t|_S})\|_\square$$

$$\leq 2\epsilon L + \frac{8L}{k^{1/4}} + \mathbf{E}_S\|W^t_\mathcal{P} - \pi(\widehat{W^t_\mathcal{P}|_S})\|_\square. \tag{2}$$

Thus, we are left with the problem of sampling from $\mathcal{P}$. Let $S = \{x_1, \ldots, x_k\}$ be a sequence of independent random variables that are uniformly distributed in $[0,1]$, and let $Z_i$ be the number of points $x_j$ that fall into the set $V_i$. It is easy to compute that

$$\mathbf{E}[Z_i] = \frac{k}{p} \quad \text{and} \quad \mathbf{Var}[Z_i] = \left(\frac{1}{p} - \frac{1}{p^2}\right)k < \frac{k}{p}.$$

The partition $\mathcal{P}'$ of $[0,1]$ is constructed into the sets $V'_1, \ldots, V'_p$ such that $\lambda(V'_i) = Z_i/k$ and $\lambda(V_i \cap V'_i) = \min(1/p, Z_i/k)$. For each $t \in [T]$, we construct the dikernel $\overline{W}^t : [0,1] \to \mathbb{R}$ such that the value of $\overline{W}^t$ on $V'_i \times V'_j$ is the same as the value of $W^t_\mathcal{P}$ on $V_i \times V_j$. Then, $\overline{W}^t$ agrees with $W^t_\mathcal{P}$ on the set $Q = \bigcup_{i,j \in [p]} (V_i \cap V'_i) \times (V_j \cap V'_j)$. Then, there exists a bijection $\pi$ such that $\pi(\widehat{W^t_\mathcal{P}|_S}) = \overline{W}^t$ for each $t \in [T]$. Then, for every $t \in [T]$, we have

$$\|W^t_\mathcal{P} - \pi(\widehat{W^t_\mathcal{P}|_S})\|_\square = \|W^t_\mathcal{P} - \overline{W}^t\|_\square \leq \|W^t_\mathcal{P} - \overline{W}^t\|_1 \leq 2L(1 - \lambda(Q))$$

$$= 2L\left(1 - \left(\sum_{i \in [p]} \min\left(\frac{1}{p}, \frac{Z_i}{k}\right)\right)^2\right) \leq 4L\left(1 - \sum_{i \in [p]} \min\left(\frac{1}{p}, \frac{Z_i}{k}\right)\right)$$

$$= 2L \sum_{i \in [p]} \left|\frac{1}{p} - \frac{Z_i}{k}\right| \leq 2L\left(p \sum_{i \in [p]} \left(\frac{1}{p} - \frac{Z_i}{k}\right)^2\right)^{1/2},$$

which we rewrite as

$$\|W^t_\mathcal{P} - \pi(\widehat{W^t_\mathcal{P}|_S})\|_\square^2 \leq 4L^2 p \sum_{i \in [p]} \left(\frac{1}{p} - \frac{Z_i}{k}\right)^2.$$

The expectation of the right hand side is $(4L^2 p/k^2) \sum_{i \in [p]} \mathbf{Var}(Z_i) < 4L^2 p/k$. By the Cauchy-Schwartz inequality, $\mathbf{E}\|W^t_\mathcal{P} - \pi(\widehat{W^t_\mathcal{P}|_S})\|_\square \leq 2L\sqrt{p/k}$.

Inserted this into (2), we obtain

$$\mathbf{E}\|W^t - \pi(\widehat{W^t|_S})\|_\square \leq 2\epsilon L + \frac{8L}{k^{1/4}} + 2L\sqrt{\frac{p}{k}} \leq 2\epsilon L + \frac{8L}{k^{1/4}} + \frac{2L}{k^{1/2}} 2^{CT/\epsilon^2}.$$

Choosing $\epsilon = \sqrt{CT/(\log_2 k^{1/4})} = \sqrt{4CT/(\log_2 k)}$, we obtain the upper bound

$$\mathbf{E}\|W^t - \pi(\widehat{W^t|_S})\|_\square \leq 2L\sqrt{\frac{4CT}{\log_2 k}} + \frac{8L}{k^{1/4}} + \frac{2L}{k^{1/4}} = O\left(L\sqrt{\frac{T}{\log_2 k}}\right).$$

Observing that $\|W^t - \pi(\widehat{W^t|_S})\|_\square$ changes by at most $O(L/k)$ if one element in $S$ changes, we apply Azuma's inequality with $\lambda = k\sqrt{T/\log_2 k}$ and the union bound to complete the proof. $\qquad\square$

The proof of Theorem 3.1 is immediately follows from Lemmas 3.2 and 3.7.

## 4 Analysis of Algorithm 1

In this section, we analyze Algorithm 1. Because we want to use dikernels for the analysis, we introduce a continuous version of $p_{n,A,\boldsymbol{d},\boldsymbol{b}}$ (recall (1)). The real-valued function $P_{n,A,\boldsymbol{d},\boldsymbol{b}}$ on the functions $f : [0,1] \to \mathbb{R}$ is defined as

$$P_{n,A,\boldsymbol{d},\boldsymbol{b}}(f) = \langle f, \widehat{A}f \rangle + \langle f^2, \widehat{\boldsymbol{d}\mathbf{1}^\top}1 \rangle + \langle f, \widehat{\boldsymbol{b}\mathbf{1}^\top}1 \rangle,$$

where $f^2 : [0,1] \to \mathbb{R}$ is a function such that $f^2(x) = f(x)^2$ for every $x \in [0,1]$ and $1 : [0,1] \to \mathbb{R}$ is the constant function that has a value of $1$ everywhere. The following lemma states that the minimizations of $p_{n,A,\boldsymbol{d},\boldsymbol{b}}$ and $P_{n,A,\boldsymbol{d},\boldsymbol{b}}$ are equivalent:

**Lemma 4.1.** *Let $A \in \mathbb{R}^{n \times n}$ be a matrix and $\boldsymbol{d}, \boldsymbol{b} \in \mathbb{R}^{n \times n}$ be vectors. Then, we have*

$$\min_{\boldsymbol{v} \in [-K,K]^n} p_{n,A,\boldsymbol{d},\boldsymbol{b}}(\boldsymbol{v}) = n^2 \cdot \inf_{f:[0,1] \to [-K,K]} P_{n,A,\boldsymbol{d},\boldsymbol{b}}(f).$$

*for any $K > 0$.*

*Proof.* First, we show that $n^2 \cdot \inf_{f:[0,1] \to [-K,K]} P_{n,A,\boldsymbol{d},\boldsymbol{b}}(f) \le \min_{\boldsymbol{v} \in [-K,K]^n} p_{n,A,\boldsymbol{d},\boldsymbol{b}}(\boldsymbol{v})$. Given a vector $\boldsymbol{v} \in [-K,K]^n$, we define $f : [0,1] \to [-K,K]$ as $f(x) = v_{i_n(x)}$. Then,

$$\langle f, \widehat{A}f \rangle = \sum_{i,j \in [n]} \int_{I_i} \int_{I_j} A_{ij} f(x) f(y) \mathrm{d}x \mathrm{d}y = \frac{1}{n^2} \sum_{i,j \in [n]} A_{ij} v_i v_j = \frac{1}{n^2} \langle \boldsymbol{v}, A\boldsymbol{v} \rangle,$$

$$\langle f^2, \widehat{\boldsymbol{d}\mathbf{1}^\top}1 \rangle = \sum_{i,j \in [n]} \int_{I_i} \int_{I_j} d_i f(x)^2 \mathrm{d}x \mathrm{d}y = \sum_{i \in [n]} \int_{I_i} d_i f(x)^2 \mathrm{d}x = \frac{1}{n} \sum_{i \in [n]} d_i v_i^2 = \frac{1}{n} \langle \boldsymbol{v}, \mathrm{diag}(\boldsymbol{d})\boldsymbol{v} \rangle,$$

$$\langle f, \widehat{\boldsymbol{b}\mathbf{1}^\top}1 \rangle = \sum_{i,j \in [n]} \int_{I_i} \int_{I_j} b_i f(x) \mathrm{d}x \mathrm{d}y = \sum_{i \in [n]} \int_{I_i} b_i f(x) \mathrm{d}x = \frac{1}{n} \sum_{i \in [n]} b_i v_i = \frac{1}{n} \langle \boldsymbol{v}, \boldsymbol{b} \rangle.$$

Then, we have $n^2 P_{n,A,\boldsymbol{d},\boldsymbol{b}}(f) \le p_{n,A,\boldsymbol{d},\boldsymbol{b}}(\boldsymbol{v})$.

Next, we show that $\min_{\boldsymbol{v} \in [-K,K]^n} p_{n,A,\boldsymbol{d},\boldsymbol{b}}(\boldsymbol{v}) \le n^2 \cdot \inf_{f:[0,1] \to [-K,K]} P_{n,A,\boldsymbol{d},\boldsymbol{b}}(f)$. Let $f : [0,1] \to [-K,K]$ be a measurable function. Then, for $x \in [0,1]$, we have

$$\frac{\partial P_{n,A,\boldsymbol{d},\boldsymbol{b}}(f(x))}{\partial f(x)} = \sum_{i \in [n]} \int_{I_i} A_{i i_n(x)} f(y) \mathrm{d}y + \sum_{j \in [n]} \int_{I_j} A_{i_n(x) j} f(y) \mathrm{d}y + 2 d_{i_n(x)} f(x) + b_{i_n(x)}.$$

Note that the form of this partial derivative only depends on $i_n(x)$; hence, in the optimal solution $f^* : [0,1] \to [-K,K]$, we can assume $f^*(x) = f^*(y)$ if $i_n(x) = i_n(y)$. In other words, $f^*$ is constant on each of the intervals $I_1, \ldots, I_n$. For such $f^*$, we define the vector $\boldsymbol{v} \in \mathbb{R}^n$ as $v_i = f^*(x)$, where $x \in [0,1]$ is any element in $I_i$. Then, we have

$$\langle \boldsymbol{v}, A\boldsymbol{v} \rangle = \sum_{i,j \in [n]} A_{ij} v_i v_j = n^2 \sum_{i,j \in [n]} \int_{I_i} \int_{I_j} A_{ij} f^*(x) f^*(y) \mathrm{d}x \mathrm{d}y = n^2 \langle f^*, \widehat{A}f^* \rangle,$$

$$\langle \boldsymbol{v}, \mathrm{diag}(\boldsymbol{d})\boldsymbol{v} \rangle = \sum_{i \in [n]} d_i v_i^2 = n \sum_{i \in [n]} \int_{I_i} d_i f^*(x)^2 \mathrm{d}x = n \langle (f^*)^2, \widehat{\boldsymbol{d}\mathbf{1}^T}1 \rangle,$$

$$\langle \boldsymbol{v}, \boldsymbol{b} \rangle = \sum_{i \in [n]} b_i v_i = n \sum_{i \in [n]} \int_{I_i} b_i f^*(x) \mathrm{d}x = n \langle f^*, \widehat{\boldsymbol{b}\mathbf{1}^T}1 \rangle.$$

Finally, we have $p_{n,A,\boldsymbol{d},\boldsymbol{b}}(\boldsymbol{v}) \le n^2 P_{n,A,\boldsymbol{d},\boldsymbol{b}}(f^*)$. $\qquad\square$

Now we show that Algorithm 1 well-approximates the optimal value of (1) in the following sense:

**Theorem 4.2.** *Let $\boldsymbol{v}^*$ and $z^*$ be an optimal solution and the optimal value, respectively, of problem* (1). *By choosing $k(\epsilon, \delta) = 2^{\Theta(1/\epsilon^2)} + \Theta(\log \frac{1}{\delta} \log\log \frac{1}{\delta})$, with a probability of at least $1 - \delta$, a sequence $S$ of $k$ indices independently and uniformly sampled from $[n]$ satisfies the following: Let $\tilde{\boldsymbol{v}}^*$ and $\tilde{z}^*$ be an optimal solution and the optimal value, respectively, of the problem $\min_{\boldsymbol{v} \in \mathbb{R}^k} p_{k, A|_S, \boldsymbol{d}|_S, \boldsymbol{b}|_S}(\boldsymbol{v})$. Then, we have*

$$\left| \frac{n^2}{k^2} \tilde{z}^* - z^* \right| \leq \epsilon L K^2 n^2,$$

*where $K = \max\{\max_{i \in [n]} |v_i^*|, \max_{i \in [n]} |\tilde{v}_i^*|\}$ and $L = \max\{\max_{i,j} |A_{ij}|, \max_i |d_i|, \max_i |b_i|\}$.*

*Proof.* We instantiate Theorem 3.1 with $k = 2^{\Theta(1/\epsilon^2)} + \Theta(\log \frac{1}{\delta} \log\log \frac{1}{\delta})$ and the dikernels $\widehat{A}$, $\widehat{\boldsymbol{d}\boldsymbol{1}^\top}$, and $\widehat{\boldsymbol{b}\boldsymbol{1}^\top}$. Then, with a probability of at least $1 - \delta$, there exists a measure preserving bijection $\pi : [0, 1] \to [0, 1]$ such that

$$\max\left\{ |\langle f, (\widehat{A} - \pi(\widehat{A|_S}))f\rangle|, |\langle f^2, (\widehat{\boldsymbol{d}\boldsymbol{1}^\top} - \pi(\widehat{\boldsymbol{d}\boldsymbol{1}^\top|_S}))1\rangle|, |\langle f, (\widehat{\boldsymbol{b}\boldsymbol{1}^\top} - \pi(\widehat{\boldsymbol{b}\boldsymbol{1}^\top|_S}))1\rangle| \right\} \leq \frac{\epsilon L K^2}{3}$$

for any function $f : [0, 1] \to [-K, K]$. Then, we have

$$
\begin{aligned}
\tilde{z}^* &= \min_{\boldsymbol{v} \in \mathbb{R}^k} p_{k, A|_S, \boldsymbol{d}|_S, \boldsymbol{b}|_S}(\boldsymbol{v}) = \min_{\boldsymbol{v} \in [-K, K]^k} p_{k, A|_S, \boldsymbol{d}|_S, \boldsymbol{b}|_S}(\boldsymbol{v}) \\
&= k^2 \cdot \inf_{f:[0,1] \to [-K, K]} P_{k, A|_S, \boldsymbol{d}|_S, \boldsymbol{b}|_S}(f) \qquad\qquad\qquad\qquad \text{(By Lemma 4.1)} \\
&= k^2 \cdot \inf_{f:[0,1] \to [-K, K]} \Big( \langle f, (\pi(\widehat{A|_S}) - \widehat{A})f\rangle + \langle f, \widehat{A}f\rangle + \langle f^2, (\pi(\widehat{\boldsymbol{d}\boldsymbol{1}^\top|_S}) - \widehat{\boldsymbol{d}\boldsymbol{1}^\top})1\rangle + \\
&\qquad\qquad\qquad\qquad \langle f^2, \widehat{\boldsymbol{d}\boldsymbol{1}^\top}1\rangle + \langle f, (\pi(\widehat{\boldsymbol{b}\boldsymbol{1}^\top|_S}) - \widehat{\boldsymbol{b}\boldsymbol{1}^\top})1\rangle + \langle f, \widehat{\boldsymbol{b}\boldsymbol{1}^\top}1\rangle \Big) \\
&\leq k^2 \cdot \inf_{f:[0,1] \to [-K, K]} \Big( \langle f, \widehat{A}f\rangle + \langle f^2, \widehat{\boldsymbol{d}\boldsymbol{1}^\top}1\rangle + \langle f, \widehat{\boldsymbol{b}\boldsymbol{1}^\top}1\rangle \pm \epsilon L K^2 \Big) \\
&= \frac{k^2}{n^2} \cdot \min_{\boldsymbol{v} \in [-K, K]^n} p_{n, A, \boldsymbol{d}, \boldsymbol{b}}(\boldsymbol{v}) \pm \epsilon L K^2 k^2. \qquad\qquad\qquad \text{(By Lemma 4.1)} \\
&= \frac{k^2}{n^2} \cdot \min_{\boldsymbol{v} \in \mathbb{R}^n} p_{n, A, \boldsymbol{d}, \boldsymbol{b}}(\boldsymbol{v}) \pm \epsilon L K^2 k^2 = \frac{k^2}{n^2} z^* \pm \epsilon L K^2 k^2.
\end{aligned}
$$

Rearranging the inequality, we obtain the desired result. $\qquad\square$

We can show that $K$ is bounded when $A$ is symmetric and full rank. To see this, we first note that we can assume $A + n\mathrm{diag}(\boldsymbol{d})$ is positive-definite, as otherwise $p_{n, A, \boldsymbol{d}, \boldsymbol{b}}$ is not bounded and the problem is uninteresting. Then, for any set $S \subseteq [n]$ of $k$ indices, $(A + n\mathrm{diag}(\boldsymbol{d}))|_S$ is again positive-definite because it is a principal submatrix. Hence, we have $\boldsymbol{v}^* = (A + n\mathrm{diag}(\boldsymbol{d}))^{-1} n\boldsymbol{b}/2$ and $\tilde{\boldsymbol{v}}^* = (A|_S + n\mathrm{diag}(\boldsymbol{d}|_S))^{-1} n\boldsymbol{b}|_S/2$, which means that $K$ is bounded.

## 5 Experiments

In this section, we demonstrate the effectiveness of our method by experiment.[1] All experiments were conducted on an Amazon EC2 c3.8xlarge instance. Error bars indicate the standard deviations over ten trials with different random seeds.

**Numerical simulation** We investigated the actual relationships between $n$, $k$, and $\epsilon$. To this end, we prepared synthetic data as follows. We randomly generated inputs as $A_{ij} \sim U_{[-1,1]}$, $d_i \sim U_{[0,1]}$, and $b_i \sim U_{[-1,1]}$ for $i, j \in [n]$, where $U_{[a,b]}$ denotes the uniform distribution with the support $[a, b]$. After that, we solved (1) by using Algorithm 1 and compared it with the exact solution obtained by QP.[2] The result (Figure 1) show the approximation errors were evenly controlled regardless of $n$, which meets the error analysis (Theorem 4.2).

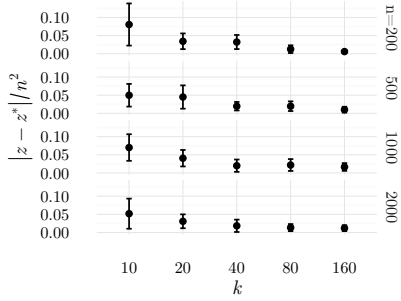

Figure 1: Numerical simulation: absolute approximation error scaled by $n^2$.

Table 1: Pearson divergence: runtime (second).

| | k | $n = 500$ | 1000 | 2000 | 5000 |
|---|---|---|---|---|---|
| Proposed | 20 | 0.002 | 0.002 | 0.002 | 0.002 |
| | 40 | 0.003 | 0.003 | 0.003 | 0.003 |
| | 80 | 0.007 | 0.007 | 0.008 | 0.008 |
| | 160 | 0.030 | 0.030 | 0.033 | 0.035 |
| Nyström | 20 | 0.005 | 0.012 | 0.046 | 0.274 |
| | 40 | 0.010 | 0.022 | 0.087 | 0.513 |
| | 80 | 0.022 | 0.049 | 0.188 | 0.942 |
| | 160 | 0.076 | 0.116 | 0.432 | 1.972 |

Table 2: Pearson divergence: absolute approximation error.

| | k | $n = 500$ | 1000 | 2000 | 5000 |
|---|---|---|---|---|---|
| Proposed | 20 | $0.0027 \pm 0.0028$ | $0.0012 \pm 0.0012$ | $0.0021 \pm 0.0019$ | $0.0016 \pm 0.0022$ |
| | 40 | $0.0018 \pm 0.0023$ | $0.0006 \pm 0.0007$ | $0.0012 \pm 0.0011$ | $0.0011 \pm 0.0020$ |
| | 80 | $0.0007 \pm 0.0008$ | $0.0004 \pm 0.0003$ | $0.0008 \pm 0.0008$ | $0.0007 \pm 0.0017$ |
| | 160 | $0.0003 \pm 0.0003$ | $0.0002 \pm 0.0001$ | $0.0003 \pm 0.0003$ | $0.0002 \pm 0.0003$ |
| Nyström | 20 | $0.3685 \pm 0.9142$ | $1.3006 \pm 2.4504$ | $3.1119 \pm 6.1464$ | $0.6989 \pm 0.9644$ |
| | 40 | $0.3549 \pm 0.6191$ | $0.4207 \pm 0.7018$ | $0.9838 \pm 1.5422$ | $0.3744 \pm 0.6655$ |
| | 80 | $0.0184 \pm 0.0192$ | $0.0398 \pm 0.0472$ | $0.2056 \pm 0.2725$ | $0.5705 \pm 0.7918$ |
| | 160 | $0.0143 \pm 0.0209$ | $0.0348 \pm 0.0541$ | $0.0585 \pm 0.1112$ | $0.0254 \pm 0.0285$ |

**Application to kernel methods**   Next, we considered the kernel approximation of the Pearson divergence [21]. The problem is defined as follows. Suppose we have the two different data sets $\boldsymbol{x} = (x_1, \ldots, x_n) \in \mathbb{R}^n$ and $\boldsymbol{x}' = (x'_1, \ldots, x'_{n'}) \in \mathbb{R}^{n'}$ where $n, n' \in \mathbb{N}$. Let $H \in \mathbb{R}^{n \times n}$ be a gram matrix such that $H_{l,m} = \frac{\alpha}{n} \sum_{i=1}^{n} \phi(x_i, x_l)\phi(x_i, x_m) + \frac{1-\alpha}{n'} \sum_{j=1}^{n'} \phi(x'_j, x_l)\phi(x'_j, x_m)$, where $\phi(\cdot, \cdot)$ is a kernel function and $\alpha \in (0, 1)$ is a parameter. Also, let $\boldsymbol{h} \in \mathbb{R}^n$ be a vector such that $h_l = \frac{1}{n} \sum_{i=1}^{n} \phi(x_i, x_l)$. Then, an estimator of the $\alpha$-relative Pearson divergence between the distributions of $\boldsymbol{x}$ and $\boldsymbol{x}'$ is obtained by $-\frac{1}{2} - \min_{\boldsymbol{v} \in \mathbb{R}^n} \frac{1}{2}\langle \boldsymbol{v}, H\boldsymbol{v} \rangle - \langle \boldsymbol{h}, \boldsymbol{v} \rangle + \frac{\lambda}{2}\langle \boldsymbol{v}, \boldsymbol{v} \rangle$. Here, $\lambda > 0$ is a regularization parameter. In this experiment, we used the Gaussian kernel $\phi(x, y) = \exp((x-y)^2/2\sigma^2)$ and set $n' = 200$ and $\alpha = 0.5$; $\sigma^2$ and $\lambda$ were chosen by 5-fold cross-validation as suggested in [21]. We randomly generated the data sets as $x_i \sim N(1, 0.5)$ for $i \in [n]$ and $x'_j \sim N(1.5, 0.5)$ for $j \in [n']$ where $N(\mu, \sigma^2)$ denotes the Gaussian distribution with mean $\mu$ and variance $\sigma^2$.

We encoded this problem into (1) by setting $A = \frac{1}{2}H$, $\boldsymbol{b} = -\boldsymbol{h}$, and $\boldsymbol{d} = \frac{\lambda}{2n}\mathbf{1}_n$, where $\mathbf{1}_n$ denotes the $n$-dimensional vector whose elements are all one. After that, given $k$, we computed the second step of Algorithm 1 with the pseudoinverse of $A|_S + k\mathrm{diag}(\boldsymbol{d}|_S)$. Absolute approximation errors and runtimes were compared with Nyström's method whose approximated rank was set to $k$. In terms of accuracy, our method clearly outperformed Nyström's method (Table 2). In addition, the runtimes of our method were nearly constant, whereas the runtimes of Nyström's method grew linearly in $k$ (Table 1).

# 6   Acknowledgments

We would like to thank Makoto Yamada for suggesting a motivating problem of our method. K. H. is supported by MEXT KAKENHI 15K16055. Y. Y. is supported by MEXT Grant-in-Aid for Scientific Research on Innovative Areas (No. 24106001), JST, CREST, Foundations of Innovative Algorithms for Big Data, and JST, ERATO, Kawarabayashi Large Graph Project.

## Footnotes

[1] The program codes are available at https://github.com/hayasick/CTOQ.

[2] We used GLPK (https://www.gnu.org/software/glpk/) for the QP solver.

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
