[Reviews · NeurIPS 2016]

Reviewer 1

Summary

This paper shows that if you take any quadratic function on $n$ variables, and restrict to a random subset of $f(eps)$ variables (and rescale), the minimum remains within eps * n^2 of the original minimum (if all the variables/terms are O(1)). As a practical matter, this is not very useful: f is exponential, and having additive error proportional to the maximal possible function value isn't very good. But it's very interesting from a theoretical perspective.

Qualitative Assessment

The experiments suggest that the algorithm is better than the theorems show, and has error more like 1/k than 1/sqrt{log k}. On the one hand, that means the algorithm is more interesting than claimed; on the other, it suggests that the given proof is the wrong approach. ---- Response to rebuttal ---- The proposed proof of Lemma 3.2 is somewhat flawed, since it seems to replace tau_1 and tau_2 by their maximum value K, but the coefficient may be negative. I think this can be fixed with a triangle inequality. In addition, the proof is not very clear: we see that the submitted 3.2 proof is incorrect, since it doesn't scale properly with K, and it's not obvious why the new proof is better.

Confidence in this Review

1-Less confident (might not have understood significant parts)


Reviewer 2

Summary

The paper proposes a sampling-based algorithm for minimizing quadratic function without constraints. The paper claims and provides theoretical analysis that the time complexity is constant while the approximation solution is bounded. Empirical evaluation has been conducted to demonstrate the advantages of the proposed sub-sampling algorithms.

Qualitative Assessment

Presentation: The written of the paper is quiet clear. The reviewer can follow the paper easily. Motivation: The idea of adopting sampling based algorithm to speed up finding the optimal solution of quadratic functions minimization makes sense. Technical contribution and significance: First, the theoretical results need to bound the function f and g in an unknown range K, and all the results have such restriction. However, the paper does not discuss K. If K = inf, the results make no sense. Although empirical evaluation shows the good performance of the proposed algorithm, this issue is not well addressed. Second, the main technical contribution is to apply the theoretical results of Theorem 3.1 to the case of Quadratic Function Minimization. The reviewer does not observe much significant theoretical gain. The reviewer would like to see results about Quadratic Function minimization with constraints. Third, the paper does not address the difference of the proposed algorithm with the Nystrom method and some recently proposed algorithms. Empirical evaluation: First, the size of empirical evaluation is small. The reviewer doubts why they need to conduct on an Amazon EC2 c3.8 large instance. Second, in Numerical simulation, what is \epsilon is unknown. Actually, from assumption in Theorem 4.2, it seems that \epsilon would affect the selection of k significantly. As shown in Theorem 4.2, the difference of estimated solution and true solution is bounded linearly with \epsilon, but the selected k should be 2^{1/\epsilon^2}. Results shown in Figure 1 seem not match the theoretical results in Theorem 4.2. The following are some insufficient parts. 1. Lack of discussion on K 2. Lack of discussion on the proposed algorithm and recent proposed algorithms, e.g., Nystrom method, etc. 3. Seem inconsistent with the results in Fig. 1 and Theorem 4.2

Confidence in this Review

2-Confident (read it all; understood it all reasonably well)


Reviewer 3

Summary

The paper considers the problem of approximating the VALUE of the optimization problem min_v < v,Av > + n* < v,diag(d)v > + n* < b,v > where A is an n x n matrix. Note that this problem has a closed form solution so the problem is basically approximating - n^2/8 * b^T (A+A^T+2*n*diag(d))^{-1} b attained at v = - n/4 * (A+A^T+2*n*diag(d))^{-1} b. The technique of the paper is to apply tools from graph limits to show that by sampling a small subset S of k coordinates from {1,..,n} and solving the new problem with A' = A restricted to rows and columns in S, d' = d restricted to coordinates in S, and b'= b restricted to coordinates in S, the solution value to the new problem is a good approximation for the original solution value (after scaling appropriately).

Qualitative Assessment

The paper is quite nice and the experimental result is promising even though it is only done on a synthetic dataset. In the experiment, the approximation quality of the new algorithm seems much better than the comparison, though it would be helpful to provide some description of the Nystrom method being considered. The proof uses interesting tools from graph limits, which is great but it also leads to rather high bounds on the number of samples (2^{1/epsilon} samples), which is necessary for general settings in graph limits but it is not clear if it is required here (in fact the examples are used to show it should be much lower in practice!). One problem is that the paper is largely theoretical and the theoretical guarantee is somewhat unsatisfactory. The main theorem is theorem 4.2 and it states that the approximation error is bounded by epsilon*L*K^2*n^2. Unfortunately K is defined based on the output of the algorithm and not an innate quantity to the original problem. This makes the guarantee weak. Even if one believes K is small, the error is proportionate to the maximum value and could be rather large.

Confidence in this Review

2-Confident (read it all; understood it all reasonably well)


Reviewer 4

Summary

The authors prove that it is possible to approximately compute the minimum of quadratic functions by solving a sub-sampled instance in a constant number of coordinates. Their main tool is a proving a weak-regularity lemma sort of result for quadratic functions. They utilize tools developed for defining limits of dense graphs. The authors present evidence for the method by solving several synthetic instances, and also apply the algorithm to quickly estimate Pearson divergence in several (synthetic) instances.

Qualitative Assessment

The main novelty of the paper is to in fact _prove_ that as long as the quadratic minimization problem has bounded coefficients, solving a subsampled instance can provide a meaningful approximation. The statement / proof is technically challenging, even though a lot of the tools were already developed in the context of graph limits. I am of the opinion that the paper needs to do more to convince that this can in fact be practically useful. Theoretically, the requirement for number of partitions (and hence samples) to be 2^(1/eps^2) is necessary (see Bounds for graph regularity and removal lemmas, Conlon and Fox). Such a high sample complexity would render the result not useful in practice. The authors claim that their experiments suggest that much smaller sample complexities suffice. However, to me, the (synthetic) experiments seem unconvincing. The instances are drawn from a large random distribution. For these instances, I suspect that concentration of measure phenomenon already implies that the optimas are tightly concentrated. The authors should at least report the range of optima and the range of answers generated by the algorithm, and hopefully design more experiments that can demonstrate practical impact. Minor comments / typos: - Lem 3.2 — There is something incorrect in the claim here. The dependence on K should be K^2 because of linearity, as can be seen by applying the lemma to f / K and g/K. I suspect the first line has a mistake. An alternate proof wouldThe proof could integrate over L = {x | h(x) <= tau}, whereby there would be no tau_1, tau_2 in the integrand. - Cor 3.4 — The second last claim ‘Since P^t is a refinement of the …,’ is the key component of the proof, and needs a proof / reference. - Lem 3.5 — The lhs should also have a factor L

Confidence in this Review

2-Confident (read it all; understood it all reasonably well)


Reviewer 5

Summary

The authors suggest a stochastic approach to approximate the minimum value of a quadratic function with $n$ variables in constant time. Their idea is to randomly select a subset of variables and minimize the quadratic function restricted to the selected subset of variables. The entire process is repeated for $k$ times, where $k$ is independent of $n$ if the relative error $\Theta(n^2\epsilon)$ is used as the stopping criteria.

Qualitative Assessment

1. Approximating the minimum value of an objective function is less interesting. In machine learning society, looking for the values of variables that minimizes the objective function is much more frequently used. 2. The proposed algorithm looks almost identical to the stochastic block coordinate descent algorithm. The differences between them has to be clarified. For more details about stochastic block coordinate descent algorithm as well its theoretical properties, please refer to [1]. 3. Although relative error may be used as a stopping criteria in practice, the absolute error $\Theta(n^2\epsilon)$ is more frequently used in deriving the theoretical complexity of an optimization algorithm. For example, [1] showed that the stochastic coordinate descent method converges linearly with absolute error. 4. The relative error should be $\Theta(n\epsilon)$ since the objective function consists of $n$ variables. [1] Y. Nesterov (2012), Efficiency of Coordinate Descent Methods on Huge-Scale Optimization Problems. SIAM Journal on Optimization.

Confidence in this Review

2-Confident (read it all; understood it all reasonably well)


Reviewer 6

Summary

This paper presents an algorithm to minimize quadratic functions in constant time, independent of the dimensionality of the problem. The algorithm works by uniformly sampling sub-dimensions from the problem and minimizing the associated quadratic functions. The paper also analyzes the properties of the proposed algorithm, and specifies the number samples required to approximate the solution. Empirical evidence of the effectiveness of the proposed approach in also provided in terms of numerical experiments.

Qualitative Assessment

Analysis of the problem from a geometrical perspective would have helped with understanding the solution better. Exploring the relation between the number of samples required and the rank of the matrix is something I would like to see.

Confidence in this Review

2-Confident (read it all; understood it all reasonably well)